# Comparison of Preclinical Properties of Several Available Antivenoms in the Search for Effective Treatment of *Vipera ammodytes* and *Vipera berus* Envenoming

**DOI:** 10.3390/toxins13030211

**Published:** 2021-03-13

**Authors:** Tihana Kurtović, Maja Lang Balija, Miran Brvar, Mojca Dobaja Borak, Sanja Mateljak Lukačević, Beata Halassy

**Affiliations:** 1Centre for Research and Knowledge Transfer in Biotechnology, University of Zagreb, Rockefellerova 10, 10000 Zagreb, Croatia; maja.langbalija@gmail.com (M.L.B.); sanjamalu@gmail.com (S.M.L.); 2Centre of Excellence for Virus Immunology and Vaccines, CERVirVac, Rockefellerova 10, 10000 Zagreb, Croatia; 3Centre for Clinical Toxicology and Pharmacology, University Medical Centre Ljubljana, Zaloška Cesta 7, 1000 Ljubljana, Slovenia; miran.brvar@kclj.si (M.B.); mojca.dobaja@kclj.si (M.D.B.); 4Centre for Clinical Physiology, Faculty of Medicine, University of Ljubljana, Zaloška Cesta 4, 1000 Ljubljana, Slovenia

**Keywords:** Zagreb antivenom, Viperfav, ViperaTAb, *Vipera ammodytes*, *Vipera berus*, preclinical efficacy

## Abstract

Snakebites are a relatively rare medical emergency in Europe. In more than half of the annual cases caused by *Vipera ammodytes*, *Vipera berus*, and *Vipera aspis*, immunotherapy with animal-derived antivenom is indicated. Among eight products recently identified as available against European medically relevant species, only Zagreb antivenom, Viperfav, and ViperaTAb have been used almost exclusively for decades. Zagreb antivenom comprises *V. ammodytes*-specific F(ab′)_2_ fragments. Viperfav is a polyspecific preparation based on F(ab′)_2_ fragments against *V. aspis*, *V. berus*, and *V. ammodytes* venoms. ViperaTAb contains Fab fragments against the venom of *V. berus*. In 2014 the production of Zagreb antivenom was discontinued. Additionally, in the period of 2017 to 2018 a shortage of Viperfav occurred. Due to a lack of the product indicated for the treatment of *V. ammodytes* bites, other antivenoms were implemented into clinical practice without comparative assessment of their eligibility. The aim of our work was to identify a high-quality antivenom that might ensure the successful treatment of *V. ammodytes* and *V. berus* bites at the preclinical level. Differentiation between bites from these two species is difficult and unreliable in clinical practice, so the availability of a unique antivenom applicable in the treatment of envenoming caused by both species would be the most advantageous for Southeastern Europe. Zagreb antivenom, Viperfav, and ViperaTAb, as well as Viper venom antitoxin for *V. berus* envenoming and the in-development Inoserp Europe, which was designed to treat envenoming caused by all medically important European snakes, were comparatively tested for the first time. Emphasis was placed on their physicochemical properties, primarily purity and aggregate content, as well as their in vivo protective efficacies. As Zagreb antivenom is no longer available on the European market, Viperfav is the highest-quality product currently available and the only antivenom whose neutralisation potency against *V. ammodytes* and *V. berus* venoms was above regulatory requirements.

## 1. Introduction

Snakebites are a relatively rare medical emergency in Europe, but their evolution may sometimes present haematological, cardiovascular, neurological, or local surgical complications [1]. An average of around 7500 cases are reported annually, with more than half representing moderate-to-severe envenoming injuries (grades 2 and 3) indicated for immunotherapy with animal-derived antivenoms. Treatment aims to prevent the onset of severe effects, particularly the swelling and spread of haemorrhage that can cause long-term disability, and to reduce the length of hospital stay [2].

*Vipera ammodytes*, *Vipera berus*, and *Vipera aspis* are venomous snakes that present the greatest public health problem in Southeastern, Southwestern, and Northern Europe, respectively [3]. In 2017, eight antivenoms against their bites were identified as available [2]. None are licensed by the European Medicines Agency. According to the systematic review of articles relating to anti-*Vipera* spp. antivenoms in Europe performed by Lamb et al. [2], about 95% of all reported cases were treated with the European viper venom antiserum (Zagreb antivenom, Croatia), Viperfav (France), or ViperaTAb (UK), which differ in several aspects. Zagreb antivenom comprises equine *V. ammodytes*-specific F(ab′)_2_ fragments whose paraspecific effectiveness against the venoms of several other medically important European snakes has been demonstrated through continuous use over 40 years in the treatment of envenoming induced by *V. aspis* (Italy) [4] and *V. berus* (UK, Sweden) [5,6,7], as well as *Macrovipera lebetina* and *Montivipera xanthina* (Turkey) [8]. Viperfav is a polyspecific preparation based on F(ab′)_2_ fragments of equine origin against *V. aspis*, *V. berus*, and *V. ammodytes* venoms. It has been authorised for use in France since 1999 and is considered clinically efficient and safe [9,10,11]. ViperaTAb contains ovine, affinity-purified Fab fragments against the venom of *V. berus*. It is indicated solely for the therapy of *V. berus* snakebites and has proved to be well-suited for envenomation in the UK and Scandinavia [6,7,12,13].

In 2014, the production of Zagreb antivenom was discontinued. Additionally, in the period from 2017 to 2018, a shortage of Viperfav occurred [14]. In some countries, this situation has forced the implementation of exceptional measures to treat *V. ammodytes* bites. In Croatia, the expiry date of remaining Zagreb antivenom doses was extended, in agreement with the Croatian Agency for Medicinal Products and Medical Devices, to the end of November 2019. Slovenia and France authorised the use of ViperaTAb [14,15]. This decision was based on its immunological cross-reactivity with *V. ammodytes* and *V. aspis* venoms, as well as on published data on ViperaTAb’s ability to neutralise their lethal effects in vivo [16], although no study comparing the effectiveness of the three antivenoms in clinical settings has been previously performed. Recently, a case series of *V. ammodytes*-bitten patients in Slovenia revealed that ViperaTAb, at the recommended dose, was not effective in the treatment of severe envenoming as it did not stop local swelling and had no effect on neurological signs [15]. As further noted in France, the use of ViperaTAb was associated with a higher risk of worsening condition in patients following antivenom treatment [14]. Such findings emphasise the need for the identification of a high-quality antivenom with an appropriate specificity profile that can ensure successful treatment of *V. ammodytes* bites, irrespective of the severity of envenomation.

Our contribution toward solving this issue constitutes an unbiased characterisation of the three most-represented European antivenoms in a relative manner, with special emphasis not only on physicochemical properties, including visual appearance, presence of antimicrobial preservatives as undesirable agents in new-generation therapeutics, F(ab′)_2_/Fab monomer content, aggregate share, protein composition and content per therapeutic dose, but also, more importantly, on in vivo protective efficacy against *V. ammodytes* venom. Inspection of the accessible products’ information leaflets revealed that neutralisation potencies are not standardised, which completely disables their comparison. Antivenoms’ protective efficacies against the venom of *V. berus* were studied as well. *V. berus* is another widespread viper of high medical importance whose habitat in Southeastern Europe mostly overlaps with that of *V. ammodytes*. Differentiation between *V. berus* and *V. ammodytes* snakebites in clinical practice is difficult and unreliable, so the availability of a unique antivenom applicable in the treatment of envenoming caused by both species would be the most advantageous outcome for Southeastern Europe. Our study covered one additional antivenom as a potential candidate, Viper venom antitoxin (Poland), which was selected as the representative most similar to ViperaTAb concerning specificity, but much less present on the market and almost completely devoid of literature data associated with its biochemical and functional properties. Finally, a new product in development, the polyvalent antivenom Inoserp Europe (Mexico), which is designed to treat envenoming caused by all medically important European snakes but is not yet available on the market [17], was also comparatively evaluated. 

## 2. Results and Discussion

As requested by the good manufacturing practices for biopharmaceuticals, antivenoms must comply with identity, purity, safety, and efficacy profiles [18]. Additionally, European Pharmacopoeia [19] specifically demands that 1 mL of drug should neutralise at least 100 median lethal doses (LD_50_) of *V. ammodytes* and *V. aspis* venoms, and at least 50 LD_50_ doses of other European vipers’ venoms.

A list of antivenoms included in this study, together with the relevant data declared by manufacturers, is presented in Table 1. Quality control of the products against *Vipera* spp. envenoming began with inspection of visual appearance. All, with the exception of Inoserp Europe, were supplied as liquid formulations. Zagreb antivenom and Biomed’s Viper venom antitoxin appeared as clear solutions, either pale yellow or colourless, which corresponded with descriptions in their marketing dossiers (Figure 1). Viperfav was slightly opalescent, while ViperaTAb was characterised as cloudy with visible particulate matter. From previous years, we had at our disposal another batch of ViperaTAb (VPT 001300), which appeared identical. Further precautionary investigation should be considered since precipitation and/or aggregation might indicate loss of activity and an increased risk of adverse reactions [20]. When reconstituted with water for injection, lyophilised Inoserp Europe produced a pale yellow, transparent solution, as already reported [17].

Concerning quantitative composition, antivenoms differed in total protein concentration and amount per therapeutic dose (Table 2). Some manufacturers did not specify information about the quantity of overall proteins in their currently available and valid products’ data sheets (Table 1). According to our results, Zagreb antivenom and ViperaTAb fulfilled the specification from the package leaflet (Table 2). Only Viperfav exceeded the preferable upper limit of 100 mg mL^−1^ [3].

The purity profile of each antivenom was assessed by SDS-PAGE (Figure 2). Monomer content was quantified by size-exclusion chromatography (SEC) (Figure 3, Table 2). Viperfav, Biomed’s Viper venom antitoxin, and Inoserp Europe were almost completely pure. Zagreb antivenom and ViperaTAb appeared equally inferior, since their immunoglobulin F(ab′)_2_ or Fab fragments, respectively, constituted less than 90% of the preparation, which is the recommended lower limit for eligibility [3]. ViperaTAb should comprise nothing but therapeutically relevant *V. berus* venom-specific antibodies that are purified by affinity chromatography [21]. Therefore, peaks/bands of molecular weight lower than 40 kDa (Figure 2 and Figure 3) might correspond not to non-IgG contaminants, but to the parts of Fab fragments whose degradation may have occurred due to instability caused by unfavourable manufacturing, storage, or transport conditions prior to distribution. According to the SEC profiles, the investigated antivenoms were substantially free from aggregates (Figure 3, Table 2). Their relative abundance was below 1%, with the exception of Inoserp Europe (3.1%). Considering impurity and aggregate content, the investigated products can be classified as safe, which is in accordance with literature data reporting low incidence of undesirable effects associated with their use in clinical setting [2].

Before any antivenom is used therapeutically in humans, and prior to introducing existing products to a new geographical area, their potency should be validated by the median effective dose (ED_50_) assay that is employed not only for routine quality control, but also to assess the ability of new products to neutralise the lethal effects of venoms from relevant snakes inhabiting the region where they are going to be adopted [3]. Treatment of *V. ammodytes* envenoming in Croatia, as in many other countries in Europe, relied on Zagreb antivenom for years until its shortage occurred. Without an obvious replacement product, the way forward, from our perspective, was to collect all potentially suitable and accessible antivenoms and test their preclinical efficacy against *V. ammodytes* venom in a comparative manner. As well as those raised against the venom of interest, monospecific products indicated for the treatment of envenomation by *V. berus* exclusively, namely ViperaTAb and Biomed’s Viper venom antitoxin, were also included in the study. As demonstrated previously, the former was shown not only to be cross-reactive with *V. ammodytes* venom, but, more significantly, to neutralise its lethal toxicity equally well as Zagreb antivenom did [16]. Despite the scarcity of clinical data, prior experience from the field suggests that ViperaTAb’s efficacy might be overrated [14,15], which led us to reconsider the potencies of all antivenoms included in the study against venoms of both species, *V. ammodytes* and *V. berus* (Table 3), irrespective of their declared specificity. Not surprisingly, monospecific Zagreb antivenom had the highest protective efficacy against *V. ammodytes* venom, followed by polyspecific Viperfav, whose neutralisation potency was approximately half that of Zagreb antivenom. Both of them fulfilled the regulatory requirements of European Pharmacopoeia [19]. The superior specific activity of Zagreb antivenom against *V. ammodytes* venom implies the more appropriately designed hyperimmunisation scheme of the Croatian producer, but also corroborates the observation that in some cases the venom of a single species elicits stronger immune response than its mixture with others of different origin, possibly due to an immunosuppressive effect [22]. Due to their homologous nature, Zagreb antivenom and Viperfav were substantially more potent than heterologous ViperaTAb which, contrary to the published results [16], did not meet the minimum regulatory standard [19]. Specific activities of Viperfav and ViperaTAb were similar, giving an example of how boosting the immunochemical purity (or share of specific IgGs in the whole antibody fraction), as in ViperaTAb, can significantly improve the potency. This should not be disregarded when comparing the neutralising capacities of different preparations. Another *V. berus*-specific antivenom, Biomed’s Viper venom antitoxin, was practically ineffective against the lethality induced by *V. ammodytes* venom, since its protective efficacy was below the ED_50_ assay sensitivity, likely because of poor immunological cross-reactivity in combination with an insufficiently high concentration. According to the literature, polyspecific Inoserp Europe should be at least as effective in counteracting the toxicity of *V. ammodytes* venom as Viperfav is, if not superior [19], but in our study its in vivo effect was even weaker than that of ViperaTAb. Similarly to Viper venom antitoxin, the cause might be the inadequate concentration of the active drug. Alternatively, Inoserp Europe might appear weaker in providing protection since the venom of snakes originating from a different geographical area (Albania), which is less typical for *V. a. ammodytes*, was used for its production. With the exception of affinity-purified antivenoms such as ViperaTAb, the immunochemical purity of those manufactured by other refinement strategies usually does not exceed 40% [23]. It seems that only a high quantity of antibodies per dose can compensate for their deficiency, thereby assuring proper protection, especially in the case of polyspecific or heterologous products.

When investigated antivenoms were challenged against *V. berus* venom in vivo, the homologous products ViperaTAb and Viper venom antitoxin did not show the expected degree of protective efficacy (Table 3). ViperaTAb’s neutralisation potency was even weaker than that obtained against *V. ammodytes* venom. In 2016 and 2017 we performed testing of another batch (VPT 001300), and an approximately 15-fold higher protective efficacy against *V. berus* venom was achieved (551.1 ± 240.0 LD_50_ mL^−1^). The observed variability in ViperaTAb’s efficacy might be associated with inconsistencies in its manufacturing process, and as such calls for heightened caution before placing each new series on the market. Viper venom antitoxin had unmeasurable activity. The best protective efficacy was accomplished with the heterologous, but polyspecific, Viperfav. It was followed by Zagreb antivenom as the second best. Inoserp Europe could not compete with either of them, appearing ineffective even when tested in the highest possible dose. Results obtained by the *V. berus* venom-based lethal toxicity neutralisation assay additionally support our presumption that antivenoms formulated as low-concentration therapeutics, prepared without affinity-based extraction of venom-specific IgGs, could not be sufficiently potent to assure adequate protection, or at least not as effective as those with a much greater active drug content—especially if hyperimmune plasma with a high titre of anti-venom antibodies was used in their production.

## 3. Conclusions

A thorough preclinical analysis of the safety-related properties and efficacy of a panel of antivenoms against *V. ammodytes* and/or *V. berus* envenoming that are currently available, or in development for the European market, was performed in a comparative manner. The study revealed Viperfav as the highest-quality product, not only because of its exceptional physicochemical characteristics, but, most importantly, its ability to assure the best protection against lethal toxicity induced by the venoms of both *V. ammodytes* and *V. berus*. Of the five tested products, it was the only one present on the market whose neutralisation potency was above regulatory standards. In comparison to Zagreb antivenom, whose effectiveness against *V. ammodytes* envenoming is well-known and proved, Viperfav showed 4- to 5-fold weaker protective efficacy and had 2-times lower overall protein content, allowing us to assume that a higher dose might be needed for successful therapy outcomes in severe snakebite cases. Zagreb antivenom, although outdated, emerged as the product with the highest specific activity and acceptable purity profile. Renewal of its production would be of great value, affirming the concept of redundancy which emphasises that, in order to ensure the availability of antivenoms to public health systems, at least several products of proven safety and efficacy should be offered in every region [24]. Inoserp Europe demonstrated much weaker neutralisation potency, which could possibly be improved by formulating it with a higher concentration of the active drug. Further analysis of new production batches of the antivenoms included in the study should be performed to confirm our results. Although monitoring of effectiveness is of utmost importance in the decision-making process, the presented findings may serve as a starting point for guidance to clinicians when choosing the most appropriate antivenom for the treatment of envenoming in Southeastern Europe.

The European region, despite not being endangered in the proportions affecting many other parts of the world, urgently requires the implementation of better regulation of, and compliance with, standards for the production, distribution, and use of these valuable medicines which are irreplaceable when confronting snakebite envenoming.

## 4. Materials and Methods

### 4.1. Snake Venom, Antivenoms, Animals, and Reagents

*V. a. ammodytes* and *V. berus* venoms were collected by milking snakes kept at the Institute of Immunology Inc., Zagreb, Croatia. Zagreb antivenom (batch number 190/1) was from the Institute of Immunology Inc., Croatia. Viperfav (batch number R4A16V) and ViperaTAb (batch number VPT 001770) were from MicroPharm Ltd., UK, supplied for the treatment of envenomed patients in Slovenia. Viper venom antitoxin (batch number 0210819000) was from Biomed, Poland. Inoserp Europe (batch number 9IT08005) was from Inosan Biopharma, Mexico. Storage of all antivenom products was in accordance with the instructions provided by the manufacturers (4–8 °C).

NIH Ola/Hsd mice (18–20 g) of both sexes were provided by the Institute of Immunology Inc., Croatia, for the lethal toxicity neutralisation assay. Animal experimentation was approved by the Croatian Ministry of Agriculture, Veterinary and Food Safety Directorate (UP/I-322-01/17-01/75, permission no. 525-10/0255-17-6, date 12 December 2017). The approval was based on the positive opinion of the National Ethical Committee (EP 110/2017). The protocols for animal care and handling were in accordance with the guidelines of the Croatian Law on Animal Welfare (2017), which strictly complies with EC Directive 2010/63/EU.

Chemicals for buffers and solutions were from Kemika (Zagreb, Croatia), unless stated otherwise.

### 4.2. Protein Characterisation and Purity Profiling

The total protein concentration of each antivenom was estimated spectrophotometrically by the use of Equation (1) [25]:*γ* (mg mL^−1^) = (*A*_228.5nm_ − *A*_234.5nm_) × *f* × dilution factor,(1)
where Ehresmann’s factor (*f*) for equine IgG of 0.2553 was used [26].

SEC analysis, which was employed for the determination of F(ab′)_2_/Fab monomer content and aggregate share, was performed on a TSK-Gel G3000SWXL column (7.8 × 300 mm) (Tosoh Bioscience, Japan) with 0.1 M phosphate-sulphate running buffer, pH 6.6, at a flow rate of 1 mL min^−1^ on a Shimadzu HPLC system (Shimadzu, Japan). The samples (5 mg mL^−1^) were loaded into the column at a volume of 50 μL per run. The effluent was monitored at 280 nm. Peaks that were obtained for antimicrobial agents (*m*-cresol or phenol) were omitted from the integration-mediated measurement of the areas under those corresponding to protein constituents. The active drug’s concentration was calculated as: [(SEC-determined monomer content in percentage/100%) × *γ* (protein)](2)

SDS-PAGE analysis of each antivenom (20 μg/well) was performed on 4–12% Bis-Tris gel using a MES running buffer under nonreducing conditions in an Xcell SureLock Mini-Cell according to the manufacturer’s protocol (Invitrogen, Carlsbad, CA, USA). Staining was carried out with acidic Coomassie Brilliant Blue (CBB). The CBB R250-stained image was recorded in transillumination mode with Amersham Imager 680 (GE Healthcare, Uppsala, Sweden). 

### 4.3. ED_50_ Test

The potential for antivenoms to neutralise the lethal toxicity of *V. a. ammodytes* and *V. berus* venoms was determined in mice, as follows. Briefly, several (minimum four) two-fold serial dilutions of antivenom were preincubated with an equal volume of the venom solution containing two median lethal doses (LD_50_, the amount of dry venom in μg causing death in half of the mice population used). The immunoprecipitates were removed by centrifugation and clear supernatants were intravenously administered to mice, sorted into groups of four. Deaths were recorded 48 h later. For each antivenom, the median effective dose (ED_50_, the amount of undiluted antivenom capable of neutralising the venom’s lethal effect in 50% of the animals) was determined. The lethal toxicity neutralisation potency (*R*) was expressed as the number of LD_50_ venom doses that can be neutralised by 1 mL of undiluted antivenom and calculated by the equation *R* = (Tv −1 )/ED_50_, where Tv represents the number of LD_50_ venom doses inoculated per mouse. *R*-value was used as a measure of the protective efficacy of each antivenom. Specific neutralisation activity (LD_50_ mg^−1^) was calculated as a ratio of *R*-value and active drug (F(ab′)_2_ or Fab) concentration. The experiment was performed independently three times for each antivenom/venom combination (or, exceptionally, only two times if the antivenom appeared ineffective when tested in the highest possible dose).

## Figures and Tables

**Figure 1 toxins-13-00211-f001:**
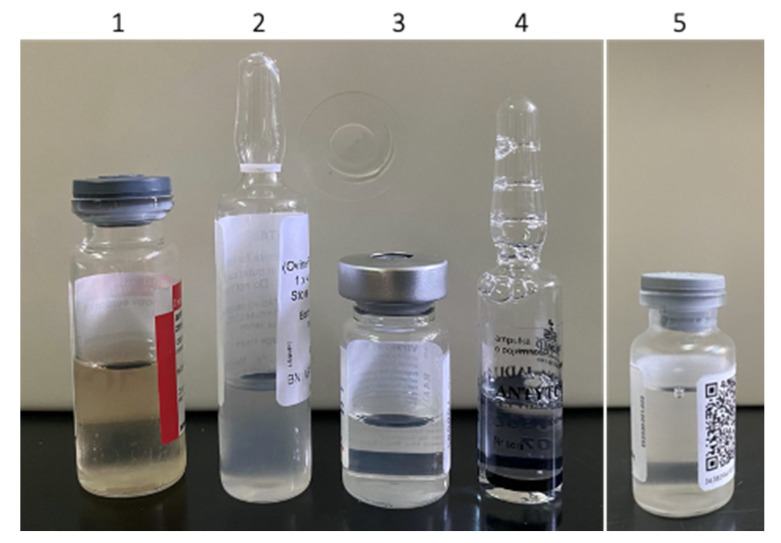
Visual appearance of Zagreb antivenom (1), ViperaTAb (2), Viperfav (3), Viper venom antitoxin (4), and Inoserp Europe (5).

**Figure 2 toxins-13-00211-f002:**
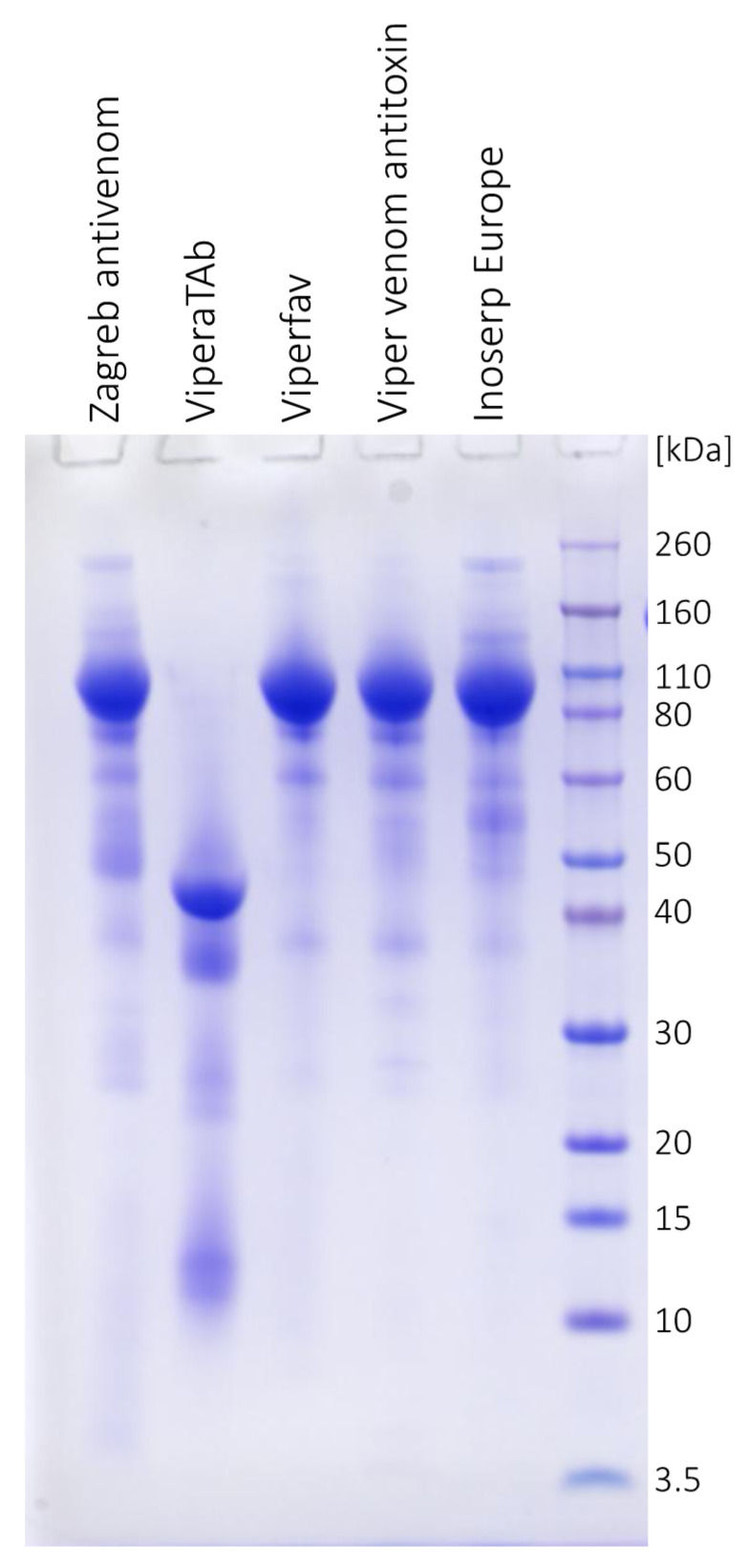
SDS-PAGE analysis of Zagreb antivenom, ViperaTAb, Viperfav, Viper venom antitoxin, and Inoserp Europe (20 μg/well) on 4–12% Bis-Tris gel under nonreducing conditions with Coomassie Brilliant Blue (CBB) R250 staining. Molecular weight standards are on the right side.

**Figure 3 toxins-13-00211-f003:**
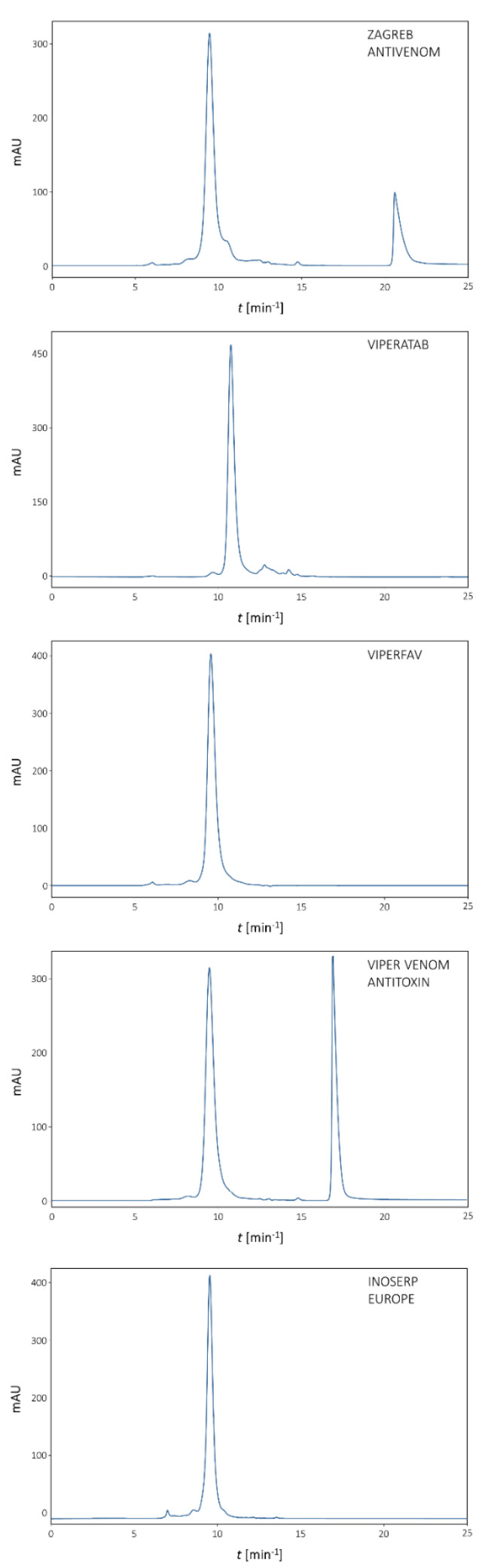
Assessment of monomer content and aggregate share by size-exclusion chromatography. The analysis was performed on a TSK-Gel G3000SWXL column (7.8 × 300 mm) with 0.1 M phosphate-sulphate running buffer, pH 6.6, at a flow rate of 1 mL min^−1^. Detection: UV at 280 nm. In Zagreb antivenom, the final eluting peak corresponds to *m*-cresol. In Viper venom antitoxin, the final eluting peak corresponds to phenol.

**Table 1 toxins-13-00211-t001:** List of antivenoms included in the study, together with data declared by manufacturers.

Product	Manufacturer	Lot Number	Exp. Date	Active Drug	Total Protein Concentration (mg mL^−1^)	Volume per Dose (mL)	Preservative	Specificity	Potency per mL
Zagreb antivenom	Institute of Immunology Inc., Croatia	190/1	11/2015	Equine F(ab′)_2_	<100	10	*m*-cresol	*V. ammodytes*	>100 ^a^
								*V. aspis*	>100
								*V. berus*	>50
								*Ma. lebetina*	>50
								*Mo. xanthina*	>50
ViperaTAb	MicroPharm Ltd., UK	VPT 001770	03/2020	Ovine Fab	25	8	none	*V. berus*	>100 ^a^
Viperfav	MicroPharm Ltd., UK	R4A16V	03/2020	Equine F(ab′)_2_	Not specified	4	none	*V. aspis*	≥250 EL. U ^b^
								*V. berus*	≥125 EL. U
								*V. ammodytes*	≥250 EL. U
Viper venom antitoxin	Biomed, Poland	210819000	07/2022	Equine F(ab′)_2_	Not specified	5	phenol	*V. berus*	>150 A.U. ^c^
Inoserp Europe	Inosan Biopharma, Mexico	9IT08005	08/2024	Equine F(ab′)_2_	17.4	10	none	*V. ammodytes*	>100 ^a^
								*V. aspis*	>100
								*V. berus* ^d^	>100

^a^ Expressed as *R* or number of median lethal doses (LD_50_) of venom that can be neutralised in vivo by one mL of antivenom; ^b^ ELISA unit; ^c^ arbitrary unit; ^d^ for full list see [17].

**Table 2 toxins-13-00211-t002:** Total protein concentration, protein quantity per dose, size-exclusion chromatography (SEC)-determined monomer content, and aggregate share of investigated antivenoms.

Product	Total Protein Concentration (mg mL^−1^) ^a^	Protein Quantity (mg/dose)	Monomer Content (%)	Aggregate Share (%)
Zagreb antivenom	99.3	993	84.3	0.6
ViperaTAb	27.9	223.2	88.1	0
Viperfav	110.6	442.4	97.3	0.9
Viper venom antitoxin	18.4	92	96.4	0.9
Inoserp Europe	20.5	205	93.8	3.1

^a^ Arithmetic mean from independent measurements.

**Table 3 toxins-13-00211-t003:** In vivo neutralisation potencies of antivenoms with their specific activities. Results are given as geometric mean from *n* independently performed experiments ± standard deviation.

	*R* (LD_50_ mL^−1^) ^a^		*R* (LD_50_/dose) ^a^		Specific Activity of Drug (LD_50_ mg^−1^) ^b^	
	*V. ammodytes*	*V. berus*	*V. ammodytes*	*V. berus*	*V. ammodytes*	*V. berus*
Zagreb antivenom	486.8 ± 31.9	131.6 ± 16.1	4868	1316	5.8	1.6
	(*n* = 2)	(*n* = 2)				
ViperaTAb	71.2 ± 34.5	38.2 ± 4.4	569.6	305.6	2.9	1.5
	(*n* = 3)	(*n* = 3)				
Viperfav	251.8 ± 62.6	192.5 ± 82.8	1007.2	770	2.3	1.8
	(*n* = 3)	(*n* = 3)				
Viper venom antitoxin	<5.1	<3.8	<25.5	<19	<0.3	<0.2
	(*n* = 2)	(*n* = 2)				
Inoserp Europe	21.6 ± 2.0	<9.0	216	<90	1.1	<0.5
	(*n* = 3)	(*n* = 2)				

^a^ Lethal toxicity neutralisation potency expressed either per one mL of the product or per therapeutic dose; ^b^ ratio of *R* to *γ*(F(ab′)_2_ or *γ*(Fab), respectively, where *γ*(active drug) was calculated as: [(SEC-determined monomer content in percentage/100%) × *γ*(protein)].

## Data Availability

Data is contained within the article.

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
