# Peer review of "Comparison of Preclinical Properties of Several Available Antivenoms in the Search for Effective Treatment of Vipera ammodytes and Vipera berus Envenoming"

_toxins, 2021, doi:10.3390/toxins13030211_

Round 1
Reviewer 1 Report
The manuscript “Comparison of preclinical properties of several available antivenoms in the search for effective treatment of Vipera ammodytes and Vipera berus envenoming” is an important evaluation of European antivenoms, especially considering the discontinuation of products and shortages. The experimental work is satisfactory; I have only minor suggestions to improve the manuscript writing.
Abstract:
This abstract is too long and needs to be better organized. I believe the abstract length for this journal is 200 words, authors should check publishing guidelines. Background information in the abstract should be reduced (such as all the details for each antivenom), and there should be more information about the specific methods and the results from the experiments. The abstract also should be organized where all antivenoms are introduced, then methods, and then results. Right now it is harder to follow when the antivenoms Viper venom antitoxin and Inoserp Europe are introduced later.
Line 25-26: Combine these two sentences.
Introduction:
Better organization is also needed in this section, try to split the two paragraphs into more focused smaller paragraphs. Introduce snakebite in Europe, the snakes (including ranges), information about snakebites from these species, and then what antivenoms are available and background on each. Finally, a paragraph on the study aims.
Results and discussion:
Again, information should be split into smaller paragraphs. These longer paragraphs make it more difficult to read. Descriptions of the physical appearances (Lines 119-128) could be added to Table 1, this would reduce text. Is it known where the venoms are sourced that are used to make the antivenoms? This information would also be useful to add to Table 1.
The results, discussion and conclusion needs to have all scientific names in italics.
Lines 208, 223 and Table 3 title: in vivo (needs to be in italics)
What about adding a western blot? It would be interesting to see what venom components might not be being neutralized between the antivenoms. If this is possible to do it would be a great addition to the study.
Methods:
I assume that ethical approval was needed for the live animal experiments? The IACUC protocol number, or similar, needs be added.
Line 319: References
Author Response
Please, see the attachment.

Reviewer 2 Report
Comparison of preclinical properties of several available anti-venoms in the search for effective treatment of Vipera ammodytes and Vipera berus envenoming
In the present review, the authors studied three most represented European antivenoms in a relative manner, with special emphasis on their physicochemical properties, and, more importantly, in vivo protective efficacy against V. ammodytes venom. In my opinion, the study is interesting and innovative, including was well delineated. However, I have some comments:
Comment (1): General proofreading is highly recommended.
Comment (2): Abstract is a little bit long and I recommend to the authors to move some of the information to the introduction. For example, take off the sentences from line 11 to 16.
- Also, the abstract missing the aim of the work although the work is very interested and important for readers and clinicians.
Comment (3): Introduction. There is a brief review of existing knowledge and relevance of study.
- Line 53: Change “that” to “which”.
Lines 54 to 57: These statements are very strange to me. How clinicians were using these antivenoms without justifications?
Comment (4): Results. Very short and missing a lot of assays to match the goal of work.
- Line 116: Put V. ammodytes and V. aspis in Italic and every others in the text.
- I am wondering Why you are studying Zagreb antivenom although its production has been discontinued. In addition, it was expired. How can you use expired product within animals? Two things, we cannot trust on the results and ethics.
- Where is the binding affinity assay?
- I recommend to the authors to include more in vitro assays to support the work.
Author Response
Please, see the attachment.

Round 2
Reviewer 2 Report
The Authors addressed all my previous comments in this improved version. I recommend to accept this MS in the present form. I also recommend to the authors to read this review article "Gutiérrez, J.M.; Vargas, M.; Segura, Á.; Herrera, M.; Villalta, M.; Solano, G.; Sánchez, A.; Herrera, C.; León, G. In Vitro Tests for Assessing the Neutralizing Ability of Snake Antivenoms: Toward the 3Rs Principles. Front Immunol 2021, 11, doi:10.3389/fimmu.2020.617429.".